# Labour pain relief practice by maternal health care providers at a tertiary facility in Kenya: An institution-based descriptive survey

**Eliazaro Gabriel Ouma, Omenge Orango, Edwin Were, Kimbley Asaso Omwodo** [ORCID] *

Department of Reproductive Health, School of Medicine, Moi University, Eldoret, Kenya

* kimbleyomwodo@alumni.harvard.edu

**Data Availability Statement:** All relevant data are within the paper and its Supporting information files.

## Abstract

### Background

Although pain relief is a crucial component of modern obstetric care, it remains a poorly established service in sub-Saharan countries such as Kenya. Maternal health care providers have an extensive role to play in meeting the analgesic needs of women during childbirth. This study sought to examine the practice of labour pain relief among Kenyan maternal health care providers.

### Methods

This was an institution-based, cross-sectional, descriptive survey. The study included midwives, obstetricians, and anaesthesiologists (n = 120) working at the second-largest tertiary facility in Kenya. A structured, self-administered questionnaire was used. The labour pain relief practice, knowledge, attitude, and perceived barriers to labour pain management were described.

### Results

One hundred and seventeen respondents participated in the study representing a response rate of 97.5%. More than half of maternal health care providers routinely provided the service of labour pain relief (61.5%). Sixty-four (88.9%) respondents reported providing pharmacological and non-pharmacological methods, while 11.1% provided only pharmacological ones. The most common pharmacological method prescribed was non-opioids (12.8%). The most preferred non-pharmacological method of pain management was touch and massage (93.8%). Regional analgesia was provided by 3.4% of the respondents. More than half of the respondents (53%) had poor knowledge of labour pain relief methods. Almost all (94%) of the respondents had a positive attitude towards providing labour pain relief. Non-availability of drugs and equipment (58.1%), lack of clear protocols and guidelines (56.4%), and absence of adequate skilled personnel (55.6%) were reported as the health system factors that hinder the provision of labour analgesia.

**Funding:** The author(s) received no specific funding for this work.

**Competing interests:** The authors have declared that no competing interests exist.

**Abbreviations:** KAP, Knowledge, attitude and practices; MTRH, Moi Teaching and Referral Hospital; RMBH, Riley Mother and Baby Hospital.

## Conclusions

More than half of maternal health care providers routinely relieve labour pain. Epidural analgesia is still relatively underutilized. There is a need to develop institutional labour pain management protocols to meet the analgesic needs of women during childbirth.

## Background

Labour pain management is a critical consideration in modern obstetrics. Although the childbirth experience is subjective and multifaceted, pain associated with labour has been described as one of the most intense forms of pain [1, 2]. Adverse consequences of labour pain may result from neuroendocrine stress responses, including increasing maternal peripheral vascular resistance and decreasing uteroplacental perfusion [3, 4]. Moreover, unrelieved labour pain has been shown to contribute to the development of postpartum psychological trauma, including postpartum depression [5].

Labour pain relief practice reflects a complex balance between the knowledge, values, and interests of maternal health care providers (MHCPs) and parturients. Both pharmacological and non-pharmacological approaches are necessary for effective labour pain management. The ideal analgesic should be safe, effective, and based on the parturients' preferences [6].

In developed economies, pain relief during labour is an integral part of intrapartum care [7], with most studies focusing on comparing the effectiveness of various methods and alternative therapies [8, 9]. However, the practice of labour pain management in Africa is suboptimal, with institutional provision reported to be as low as 13.8% [10].

Limited literature exists on maternal health care provider labour pain practice in Kenya, making it challenging to assess trends in this population. The objective of the study was to assess the labour pain relief practice by maternal healthcare providers working in the second largest public hospital in Kenya.

### Conceptual framework

Interview data were constructed by adopting the Reciprocal Determinism (Reciprocal Causation) model, which has been shown to be superior to other models in assessing pain management practices by healthcare workers [11, 12]. In this study, the internal personal factors included healthcare provider factors, demographic factors (e.g., sex, age, professional cadre, and duration of practice), knowledge and attitude. The environmental factors included the social milieu within which MHCPs continually interact, i.e., health system factors (e.g., availability of adequate skilled personnel, clear protocols and guidelines, drugs and equipment).

## Methods

### Study design

An institution-based, cross-sectional descriptive survey of labour pain relief practice was conducted through a structured, self-administered questionnaire to maternal health care providers.

This study site was the Moi Teaching and Referral Hospital (MTRH), Eldoret, Kenya. MTRH is the second-largest public teaching and referral hospital in Kenya. The Hospital serves mostly residents from the Western Kenya Region (representing at least 22 Counties), parts of Eastern Uganda and Southern Sudan with a population catchment of approximately 24 million. It is also the main teaching center for Moi University School of Medicine that trains

midwives, anaesthesiologists and obstetricians with an annual turnover of over 50 practitioners involved directly and indirectly in the management of labour and its outcomes. In the year 2019/2020, there was an average of 800 vaginal deliveries at MTRH.

Ethics approval and consent to participate Ethical clearance was obtained from the Moi University Institutional Research Ethics Committee (IREC) REF: IREC/2020/144. Permission from MRTH administration was also obtained. Written informed consent was taken from each subject before enrolment in the study following Good Clinical Practice (GCP) principles, and all methods were carried out in accordance with the Declaration of Helsinki. All data were maintained as confidential, and no individual was identified in the dissemination of findings.

## Study population

The study population comprised midwives, residents and consultants in both anaesthesia and reproductive health working at MTRH during the data collection period. The study period was between January 1st 2021, and March 31st 2021.

Obstetric caregivers (obstetricians, anaesthesiologists, residents and midwives) at MTRH, who are involved in providing labour pain relief and consented to the study, were included.

Medical officer interns, working alongside residents and consultants (who are the primary decision-makers) and with short exposure time in the maternity unit, were excluded from the survey, as they play a secondary role in patient management.

There were no labour pain management guidelines within the maternity unit at the time of the study. The service of labour pain relief was therefore by provider preference.

## Eligibility criteria

**Inclusion criteria.**   • Obstetric caregivers (obstetricians, anaesthesiologists, residents and midwives) at MTRH, who were involved in the provision of labour analgesia and consented to the study.

**Exclusion criteria.**   • Obstetric caregivers (obstetricians, anaesthesiologists, residents and midwives) at MTRH who were on leave and away from the study area during the period of the study.

• First-year residents in both Reproductive Health and Anaesthesia who having just joined their respective programs did not have sufficient exposure time in the facility at the time of the survey.

**Study size.**   The study employed a census that included all the 120 maternal health care providers of interest at MTRH.

**Selection of participants.**   Consecutive sampling method was used until all eligible participants in the study were enrolled.

**Study variables.**   Dependent variable: labour pain relief practice–choice, pattern and frequency of use of labour analgesia options as self-reported by MHCPs in a standardized questionnaire.

*Independent variables.* Internal personal factors, including MHCPs demographic factors (e.g. sex, age, professional cadre and duration of practice), knowledge, skills, perception and attitude. Socio-demographic characteristics i.e age, sex, profession, years of service, department of practice.

The environmental factors included patient related factors (e.g., patient knowledge and attitude towards labour analgesia) and the institutional related elements (e.g., availability of resources, protocols, guidelines, regulation, and ratio of professionals to patient).

## Study instrument

After obtaining written informed consent, MHCPs who met the inclusion criteria were requested to complete a paper-based structured questionnaire. The questionnaire contained 44 questions and was self-administered. The study adopted a questionnaire based on similar studies, the conceptual framework and study objectives [13, 14].

The questionnaire comprised the following sections:

Section A: sociodemographic characteristics, i.e., sex, age, professional cadre, and duration of practice. Section B assessed the provider's knowledge and attitude towards labour analgesia. A total of 10 items were included in the questionnaire to assess the respondents' knowledge, and five items were included to assess the respondents' attitudes. The reliability analysis of these ten items on knowledge was performed and found to be at an acceptable level of standardized Cronbach's alpha [$\alpha$ = .718].

Participants' overall knowledge level was categorized using a modified Bloom's cut-off point: good if the score was between 80% and 100% (12–15 points), moderate if the score was between 50% and 79% (7–11 points), and poor if the score was less than 50% ($<$7 points).

Attitudes towards labour pain relief were assessed using three-level Likert items. The response options for these items were 'disagree,' 'unsure,' and 'agree.' The reliability of these three items was acceptable with a standardized Cronbach's alpha ($\alpha$ = .804). Similarly, the attitude of healthcare providers towards the provision of labour pain relief was categorized using the original Bloom cut–off point. Attitude was considered positive if the score was 80%–100% (12–15 points), neutral if the score was 60%–79% (9–11 points) and negative if the score was less than 60% ($<$ 9 points) [15]. A positive attitude towards the provision of labour pain relief meant perceiving that labour pain is significant enough to warrant intervention and that the provision of labour pain relief should be routine and not an exception.

Section C included questions on the type and frequency of use of the various forms of labour analgesics.

Only participants who responded as providing any form of labour pain relief 'routinely' were considered to be practising the provision of labour analgesia.

Section D assessed the factors influencing the provision of labour pain relief and employed a five-level Likert scale from strongly agree to strongly disagree. The first three questions assessed the health system factors influencing the provision of labour analgesia, while the remaining nine assessed other perceived barriers to the provision of labour analgesia.

The concluding section enquired about the provider's willingness to receive further training on labour analgesia.

To assure the reliability and validity of the data, a pilot study was done. The self-administered questionnaire was pretested on 12 healthcare providers i.e., 10% of the study population. The pilot study was conducted at Jaramogi Oginga Odinga Teaching and Referral Hospital (JOOTRH), a level 5 hospital located in Kisumu County.

The healthcare providers completed the questionnaire on two separate occasions, done two weeks apart. The period of two weeks was considered long enough for participants to have forgotten their responses but not long enough for a real change to occur in their knowledge, practice or barriers experienced. A discussion ensued to select the best terms for clarity of the questions, accuracy of the knowledge measured, and interpretability.

**Table 1. Sociodemographic characteristics of maternal healthcare providers.**

| Sociodemographic characteristics | Frequency | Percentage (%) |
|---|---|---|
| **Sex** | | |
| Male | 58 | 49.6 |
| Female | 59 | 50.4 |
| **Age (years)** | | |
| ≤30 | 7 | 6 |
| 31–40 | 84 | 71.8 |
| > 40 | 26 | 14.5 |
| Min.–Max. | 27.0–60.0 | |
| Mean ± SD. | 38.44 ± 7.41 | |
| **Profession** | | |
| Anaesthesiologist | 16 | 13.7 |
| Midwife | 48 | 41 |
| Obstetrician | 53 | 45.3 |
| **Duration of practice (years)** | | |
| ≤10 | 78 | 66.7 |
| 11–20 | 35 | 29.9 |
| >20 | 4 | 3.4 |
| Min.–Max. | 2.0–26.0 | |
| Mean ± SD. | 9.54 ± 4.94 | |

Participants were not informed of the second administration of the questionnaire on the first occasion. Two sets of responses (i.e., on the first and the second administration) were used to measure test-retest reliability. The first administration's responses assessed construct validity and internal consistency reliability.

Cronbach's alpha with r = 0.7 or greater was considered sufficiently reliable. In the reliability analysis, all questions had alpha scores of 0.7 to 0.9, implying respectable to good reliability.

## Results

Of the 120 maternal health care providers approached, 117 responded, representing a 97.5% response rate. Table 1 demonstrates the sociodemographic characteristics of the participants.

### The pattern of provision of labour analgesics by maternal health care providers

Seventy-two respondents (61.5%) routinely provided the service of labour pain relief. Of these, 88.9% reported offering both pharmacological and non-pharmacological methods, while 11.1% provided only pharmacological methods. Slightly more than half of all respondents (54.7%) reported routinely providing non-pharmacological methods of labour analgesia. Non-opioids were the most common pharmacological method prescribed by 13.4% (n = 15) of the respondents. Nine (8.7%) participants reported routinely providing opioids. Regional analgesia was routinely prescribed by 3.6% (n = 4) of respondents. None of the MHCPs routinely practiced labour pain management by inhalational analgesics (Table 2).

**Anaesthesiologists—Both registrar and consultant.** Non-opioid use for routine labour pain relief was reported by 30.8% (n = 4) of the anaesthesiologists. Opioids use was by 23.1% (n = 3) of the respondents within this cadre, while regional analgesics and non-pharmacological methods of pain relief were each provided by 15.4% (n = 2) of the respondents. Inhalational

**Table 2. Maternal healthcare providers' pattern of provision of labour pain relief.**

| Pain relief method | Registrar Anaesthesiologist (n = 10) | Anaesthesiologist (n = 6) | Midwife (n = 48) | Obstetrician (n = 19) | Registrar Obstetrician (n = 34) | Total (n = 117) |
|---|---|---|---|---|---|---|
| **Opioids** | | | | | | |
| Missing | 1 | 2 | 9 | 0 | 2 | 14 |
| Never | 3 (33.3%) | 1 (25.0%) | 4 (10.3%) | 1 (5.3%) | 4 (12.5%) | 13 (12.6%) |
| Occasionally | 5 (55.6%) | 1 (25.0%) | 35 (89.7%) | 12 (63.2%) | 23 (71.9%) | 76 (73.8%) |
| On maternal request | 0 (0.0%) | 0 (0.0%) | 0 (0.0%) | 1 (5.3%) | 4 (12.5%) | 5 (4.9%) |
| Routinely | 1 (11.1%) | 2 (50.0%) | 0 (0.0%) | 5 (26.3%) | 1 (3.1%) | 9 (8.7%) |
| **Non-opioids** | | | | | | |
| Missing | 1 | 2 | 1 | 0 | 1 | 5 |
| Never | 0 (0.0%) | 1 (25.0%) | 6 (12.8%) | 3 (15.8%) | 8 (24.2%) | 18 (16.1%) |
| Occasionally | 7 (77.8%) | 1 (25.0%) | 38 (80.9%) | 10 (52.6%) | 17 (51.5%) | 73 (65.2%) |
| On maternal request | 0 (0.0%) | 0 (0.0%) | 0 (0.0%) | 1 (5.3%) | 5 (15.2%) | 6 (5.4%) |
| Routinely | 2 (22.2%) | 2 (50.0%) | 3 (6.4%) | 5 (26.3%) | 3 (9.1%) | 15 (13.4%) |
| **Inhalational** | | | | | | |
| Missing | 0 | 2 | 1 | 0 | 1 | 4 |
| Never | 10 (100.0%) | 3 (75.0%) | 47 (100.0%) | 18 (94.7%) | 33 (100.0%) | 109 (96.5%) |
| Occasionally | 0 (0.0%) | 1 (25.0%) | 0 (0.0%) | 0 (0.0%) | 0 (0.0%) | 1 (0.9%) |
| On maternal request | 0 (0.0%) | 0 (0.0%) | 0 (0.0%) | 1 (5.3%) | 0 (0.0%) | 1 (0.9%) |
| **Regional agent** | | | | | | |
| Missing | 0 | 1 | 3 | 0 | 1 | 5 |
| Never | 6 (60.0%) | 0 (0.0%) | 38 (84.4%) | 4 (21.1%) | 24 (72.7%) | 72 (64.3%) |
| Occasionally | 3 (30.0%) | 4 (80.0%) | 7 (15.6%) | 10 (52.6%) | 9 (27.3%) | 33 (29.5%) |
| On maternal request | 0 (0.0%) | 0 (0.0%) | 0 (0.0%) | 3 (15.8%) | 0 (0.0%) | 3 (2.7%) |
| Routinely | 1 (10.0%) | 1 (20.0%) | 0 (0.0%) | 2 (10.5%) | 0 (0.0%) | 4 (3.6%) |
| **Nonpharmacological** | | | | | | |
| Missing | 0 | 1 | 0 | 0 | 1 | 2 |
| Never | 1 (10.0%) | 2 (40.0%) | 1 (2.1%) | 1 (5.3%) | 3 (9.1%) | 8 (7.0%) |
| Occasionally | 6 (60.0%) | 2 (40.0%) | 11 (22.9%) | 8 (42.1%) | 12 (36.4%) | 39 (33.9%) |
| On maternal request | 2 (20.0%) | 0 (0.0%) | 0 (0.0%) | 1 (5.3%) | 1 (3.0%) | 4 (3.5%) |
| Routinely | 1 (10.0%) | 1 (20.0%) | 36 (75.0%) | 9 (47.4%) | 17 (51.5%) | 64 (54.7%) |

analgesics were not routinely provided by any of the anaesthesiologist respondents. No response was obtained from 3 of 16 respondents within the cadre.

**Midwives.** Of the 48 midwife respondents, a majority (75.0%) reported providing non-pharmacological methods for labour pain management. Non-opioids were the most routinely provided pharmacological treatment for labour pain by 6.4%. None of the midwife respondents reported routine provision of opioids or regional and inhalational methods for labour pain relief.

**Obstetrician—Both registrar and consultant.** Twenty–six (50%) obstetrician respondents reported providing non-pharmacological modes of labour pain management. Non-opioids were the primary pharmacological agents provided by the majority (15.4%) of respondents, and 11.8% (n = 6) reported providing opioids routinely. Regional analgesics were

**Table 3. Maternal healthcare providers' routine labour pain relief methods.**

| Agent | Frequency | %* |
|---|---|---|
| **Opioids** | | |
| Tramadol | 8 | 88.9 |
| Morphine | 5 | 55.6 |
| Pethidine | 3 | 33.3 |
| Fentanyl | 3 | 33.3 |
| **Reported provision of any opioid** | **9** | **7.7** |
| **Nonopioids** | | |
| Buscopan | 10 | 66.7 |
| Panadol | 10 | 66.7 |
| Diclofenac | 2 | 13.3 |
| **Reported provision of any non-opioid** | **15** | **12.8** |
| **Regional** | | |
| Epidural | 3 | 75.0 |
| Spinal | 2 | 50.0 |
| **Reported provision of any regional** | **4** | **3.4** |
| **Nonpharmacological** | | |
| Touch and massage | 60 | 93.8 |
| Deep breathing/patterned breathing (Lamaze) | 52 | 81.3 |
| Maternal movements and positional changes | 52 | 81.3 |
| Social support (Reassurance) | 51 | 79.7 |
| Audio analgesia | 24 | 37.5 |
| Yoga | 3 | 4.7 |
| Intermittent local heat and cold therapy | 1 | 1.6 |
| Acupuncture | 1 | 1.6 |
| **Reported provision of any non-pharmacological** | **64** | **54.7** |

* Percentages do not add to 100% because some respondents reported providing multiple methods for labour analgesia.

provided by 3.8% (n = 2) of the respondents, while none of the obstetrician respondents reported providing inhalational agents for labour pain management.

Cumulatively, tramadol was the most routinely provided opioid analgesic by 88.9% (n = 8) of the maternal health care providers. Buscopan and paracetamol were the most routinely (66.7%) prescribed non-opioid analgesics. Epidural analgesics were the most preferred regional analgesia by 75% (n = 3) of MHCPs. The four most routinely prescribed non-pharmacological methods for labour pain relief were touch and massage (93.8%), deep breathing/patterned breathing (Lamaze techniques) (81.3%), maternal movements and positional changes (81.3%) and social support (reassurance) (79.7%) (Table 3).

## Factors influencing the provision of labour pain relief

**Providers' knowledge.** Only 5 (4.3%) of the maternal health care providers rated as having good knowledge of labour pain relief practises. All the consultant anaesthesiologists and 52.6% of the consultant obstetricians rated moderately regarding overall knowledge of labour analgesia. The proportion of those rated as having poor knowledge of labour pain relief was higher among registrar obstetricians (70.6%). Based on the composite score of 6.7/15, a

**Table 4. Maternal healthcare providers' knowledge of labour pain relief.**

| Cadre | Good | Moderate | Poor | Average score* | Score % |
|---|---|---|---|---|---|
| Anaesthesiologist (n = 6) | 0 (0.0%) | 6 (100%) | 0 (0.0%) | **9.5** | **63.3** |
| Resident anaesthesiologist (n = 10) | 0 (0.0%) | 7 (70.0%) | 3 (30.0%) | **7.4** | **49.3** |
| Midwife (n = 48) | 1 (2.1%) | 18 (37.5%) | 29 (60.4%) | **6.2** | **41.3** |
| Obstetrician (n = 19) | 3 (15.8%) | 10 (52.6%) | 6 (31.6%) | **7.7** | **51.3** |
| Resident obstetrician (n = 34) | 1 (2.9%) | 9 (26.5%) | 24 (70.6%) | **6.1** | **40.7** |
| **TOTAL n = 117** | **5 (4.3%)** | **50 (42.7%)** | **62 (53.0%)** | **6.7** | **44.7** |

* Maximum score of 15

majority (53.0%) of maternal health care providers at MTRH had poor knowledge of labour analgesia, as assessed using the modified Blooms cut-off points (Table 4).

In the self-assessment of previous education concerning labour analgesia, 81.2% (n = 95) of the participants had a "yes" response. The reported sources of labour pain relief knowledge were as part of the curriculum in previous education (60.8%), in-service education (52.6%), literature/the internet (39.2%), and colleagues (27.8%) (Table 5).

A total of 72.6% (n = 85) of MHCPs reported being aware of the universal pain assessment tools; however, only 36.8% used these tools to assess labour pain. Of the respondents reporting the use of the universal pain assessment tools, a majority (71.4%) of the anaesthesiologist and obstetricians (38.4%) preferred using the visual component, whereas the majority (74.0%) of the midwives preferred the verbal component (Table 6). Notably, 65.8% of respondents were aware of the WHO analgesic ladder. Of these, 47.0% used this tool during labour pain management. Overall, anaesthesiologists had better knowledge of the pain assessment tools than the other cadres surveyed.

There was poor overall knowledge of opioid dose properties, with only 23.7% (n = 27) of all the respondents aware that opioids do not have a ceiling effect. More than half (58.1%) of the MHCPs were aware that non-pharmacological pain relief methods are safer than pharmacological analgesics, and 76.1% were also aware that pharmacological pain relief methods increase women's comfort in labour compared to non-pharmacological analgesics.

**Attitude.** Based on the composite score of 13.3, 88.7% of MHCPs generally had a positive attitude towards the provision of labour analgesia, as assessed using the original Bloom cut-off points (Table 7).

Forty-three (36.8%) respondents expected women to feel pain during labour. A majority (82.1%) of the respondents agreed that labour pain should be relieved, with an equal number agreeing that labour pain relief improves the overall birth experience. However, ten (8.5%) of the study subjects believed that labour is a natural process that does not require any analgesia; 17.1% were unsure, while the remaining 74.4% disagreed (Table 8).

**Table 5. Sources of knowledge on labour pain relief.**

| Cadre | As part of the curriculum in previous education | During in-service education (C.M.E, Seminars etc.) | literature / the internet | From colleagues |
|---|---|---|---|---|
| Anaesthesiologist (n = 13) | 76.9 | 69.2 | 30.8 | 7.7 |
| Midwife (n = 38) | 55.3 | 42.1 | 34.2 | 23.7 |
| Obstetrician (n = 46) | 60.9 | 56.5 | 45.7 | 36.9 |
| **Total (n = 97)** | **60.8** | **52.6** | **39.2** | **27.8** |

**Table 6. Maternal healthcare providers' use of pain assessment tools in labour pain relief.**

| Cadre | WHO analgesic ladder(n = 55) | Universal pain assessment tools | | | | Total using UPA* (N = 43) |
|---|---|---|---|---|---|---|
| | | Numerical | | Visual | Verbal | |
| Anaesthesiologist (N = 16) | 81.3 | | 42.9 | 71.4 | 28.6 | 43.8 |
| Midwife (N = 48) | 31.3 | | 21.8 | 34.8 | 74.0 | 47.9 |
| Obstetrician (N = 53) | 50.9 | | 30.1 | 38.4 | 30.8 | 24.5 |
| | | Total† | 27.9 | 41.9 | 53.5 | |
| Total (N = 117) | 47.0 | | | | | 36.8 |

† Values do not add up to 100% because some respondents reported using more than one tool.

UPA*: Universal pain assessment tools.

**Table 7. Maternal healthcare providers' attitude towards labour pain relief.**

| Cadre | Positive | Neutral | Negative | Avg Score† | Score% |
|---|---|---|---|---|---|
| Anaesthesiologist (N = 6) | 6(100.0%) | 0(0.0%) | 0(0.0%) | 13.2 | 88 |
| Resident anaesthesiologist (N = 10) | 9(90.0%) | 1(10.0%) | 0(0.0%) | 13 | 86.7 |
| Midwife (N = 48) | 45(93.8%) | 3(6.3%) | 0(0.0%) | 13.2 | 89.3 |
| Obstetrician (N = 19) | 18(94.7%) | 1(5.3%) | 0(0.0%) | 13.4 | 89.5 |
| Resident obstetrician (N = 33) | 31(93.9%) | 2(6.1%) | 0(0.0%) | 13.5 | 90 |
| TOTAL N = 116 | 109(94.0%) | 7(6.0%) | 0(0.0%) | 13.3 | 88.7 |

†Maximum score of 15

**Table 8. Maternal healthcare providers' specific attitude toward provision of labour pain relief.**

| Cadre | Percentage of maternal health care providers' who agree that: | | | Percentage of maternal health care providers' who disagree that: | |
|---|---|---|---|---|---|
| | Women are expected to feel pain during labour | Pain in labour should be relieved | Relief of Labour pain improves the overall maternal experience | Labour is a natural process that does not require analgesia | Patients complaining of pain during labour may be seeking attention |
| Anesthesiologist (N = 16) | 18.8 | 100 | 87.5 | 93.8 | 100 |
| Midwife (N = 48) | 50 | 66.7 | 70.8 | 50 | 77.1 |
| Obstetrician (N = 53) | 30.2 | 90.6 | 90.6 | 90.6 | 86.8 |
| Total (N = 117) | 36.8 | 82.1 | 82.1 | 74.4 | 84.6 |

**Health system factors.** A majority (91.7%) of maternal health care providers at MTRH reported experiencing health system factors that hindered their provision of labour analgesia. These included the non-availability of drugs and equipment (58.1%), a lack of clear protocols and guidelines (56.4%) and an absence of adequate skilled personnel (55.6%).

Other barriers/factors hindering the provision of labour pain relief included (N = 117):

1. Fear of foetal distress (47.1%)

2. Fear of adverse maternal effects (41.8%)

3. Cost implications (perceived as expensive) (36.7%)

4. Fear that it may increase the incidence of caesarean sections and instrumental delivery (34.2%)

Thirteen (11.1%) respondents reported that oftentimes, patients decline labour analgesia.

Almost all the participants (94%) reported that the introduction of labour pain relief guidelines would improve the management of labour at MTRH, while 95.7% indicated that regular courses on effective labour pain relief would be useful in their practice of labour analgesia.

## Discussion

Slightly over half of MHCPs routinely provided the service of labour pain relief (mainly non-pharmacological). This proportion may be considered inadequate considering the hospitals tertiary status and might allude to a lesser provision of labour pain relief services in lower-level facilities in Kenya. This proportion was higher than similar studies conducted in Ibadan, Nigeria and Hawassa, Ethiopia [10, 16]. This difference might be due to the inclusion of different-level public healthcare institutions in the prior studies and, consequently, a difference in the knowledge and availability of resources.

Non-pharmacological labour pain relief methods, such as massage, breathing techniques, maternal movements, positional changes, support, and companionship, were the most frequently provided, likely due to their safety profile. This approach is often the first-line option for labour pain management in many settings.

Non-opioid pharmacological analgesics, particularly Buscopan and paracetamol, were the most commonly prescribed by maternal health care providers. Opioid use was low, potentially due to poor pain scoring, concerns about side effects, and a reliance on non-pharmacological pain relief methods.

While pethidine injection was previously the most prescribed opioid for labour pain relief [17], tramadol was the most preferred opioid reported in this study, as pethidine is now a highly restricted drug at the study facility.

The World Health Organization (WHO) recommends epidural analgesia for healthy pregnant women requesting pain relief during labour, depending on a woman's preferences [18]. Data on epidurals in developing countries are scarce, but there is a generally low provision of labour epidurals [19, 20]. In this study, only 3.6% of the respondents reported offering regional analgesia routinely. Epidural analgesia is provided to approximately 30% and 73% of women in labour in the United Kingdom and the US, respectively, with increasing rates expected globally [21, 22]. In our setting, epidural analgesia is reserved for women with pregnancies complicated by medical conditions.

MHCPs' at MTRH had generally poor knowledge of labour pain relief methods, which may hinder accurate assessment, identification, recommendation, and consultation for proper pain relief. These findings were slightly higher than those reported from the Amhara region, Ethiopia, 48.5% [23], but lower than those reported from Ibadan, Nigeria, 66.7% [16]. The difference may be due to the study participant's variance in demographic characteristics. The study also found that most resident obstetricians (70.6%) and midwives (60.4%) had poor knowledge scores, highlighting potential gaps in reproductive health course content for health professionals.

In the current findings, almost all MHCPs were positive about providing labour pain relief. This is higher than similar studies in Ethiopia, 57.2% [23], which may signify a cultural disparity between the study populations.

In contrast to studies from Ethiopia and Nigeria [23, 24], this study found no significant association between the provision of labour pain relief and the knowledge or attitude of maternal health care providers.

Health system barriers such as non-availability of drugs and equipment, lack of clear protocols and guidelines, and inadequate skilled personnel were reported as significant hindrances (by 91.7%) to the use of labour pain relief methods in this study.

This finding was consistent with studies from Ethiopia [25], Saudi Arabia [26] and Nigeria [19].

## Limitations of the study

This was a tertiary institution-based study conducted in Eldoret, Kenya; hence, the conclusions can only be generalized to similar-level hospitals with equal capacity. We also recommend further studies to explore a broader scope of MHCPs perspectives to comprehensively address the cause-and-effect relationship of the factors affecting the provision of labour analgesia.

## Conclusions

Maternal health care providers regularly use non-pharmacological methods, such as massage, breathing techniques, and position changes, to relieve labour pain. Although epidural analgesia is the preferred method, it is underused, and most providers have inadequate knowledge about it. Improving education in labour pain management for healthcare professionals and establishing standard procedures in healthcare institutions could help address this knowledge gap.

Health system barriers impede the provision of labour pain analgesia. Interventions aimed at addressing these barriers are necessary for improving the quality of care for women during labour and delivery in Kenya. We recommend further studies to explore the in-depth perspectives of providers and pregnant women on labour pain management.

## Supporting information

**S1 Data.**
(XLSX)

## Acknowledgments

We appreciate the doctors, anaesthesiologists, and midwives at the Reproductive Health Department of MTRH for their enormous support in participating in the study.

## Author Contributions

**Conceptualization:** Eliazaro Gabriel Ouma, Omenge Orango, Edwin Were.

**Data curation:** Eliazaro Gabriel Ouma.

**Formal analysis:** Eliazaro Gabriel Ouma, Kimbley Asaso Omwodo.

**Investigation:** Eliazaro Gabriel Ouma.

**Methodology:** Eliazaro Gabriel Ouma, Kimbley Asaso Omwodo.

**Supervision:** Omenge Orango, Edwin Were.

**Writing – original draft:** Eliazaro Gabriel Ouma, Kimbley Asaso Omwodo.

**Writing – review & editing:** Eliazaro Gabriel Ouma, Kimbley Asaso Omwodo.

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
