## [Decision Letter · Decision Letter 0]

13 Sep 2023

PONE-D-23-09684Labour pain relief practice by maternal health care providers in Kenya: An institution-based descriptive surveyPLOS ONE

Dear Dr. Omwodo,

Thank you for submitting your manuscript to PLOS ONE. After careful consideration, we feel that it has merit but does not fully meet PLOS ONE’s publication criteria as it currently stands. Therefore, we invite you to submit a revised version of the manuscript that addresses the points raised during the review process.

We look forward to receiving your revised manuscript.

Kind regards,

Daniel Semakula, M.D. MPH, PhD

Academic Editor

PLOS ONE

Journal Requirements:

Reviewers' comments:

Reviewer's Responses to Questions

**Comments to the Author**

1. Is the manuscript technically sound, and do the data support the conclusions?

Reviewer #1: Yes

Reviewer #2: Yes

2. Has the statistical analysis been performed appropriately and rigorously? 

Reviewer #1: Yes

Reviewer #2: Yes

3. Have the authors made all data underlying the findings in their manuscript fully available?

Reviewer #1: Yes

Reviewer #2: Yes

4. Is the manuscript presented in an intelligible fashion and written in standard English?

Reviewer #1: Yes

Reviewer #2: Yes

5. Review Comments to the Author

Reviewer #1: Title: Labour pain relief practice by maternal health care providers in Kenya: An institution based

Descriptive survey

An institution-based, cross-sectional, descriptive survey study included midwives, obstetricians, and anaesthesiologists working at tertiary facility in Kenya.

1. The title: Labour pain relief practice by maternal health care providers in Kenya: An institution-based descriptive survey: You have used an institutional survey which seems acceptable being conducted in one hospital although you have identified that 800 deliveries recorded per month .At the same time you have generalized the setting to whole Kenya!! You have to convince the reader how it can represent all Kenya!!!

My advice is to fix on an institutional survey at Kenya

1. Line 85, Review and approval were obtained from the Moi University School of Medicine/MTRH Institutional Research and Ethics Committee. All participants provided informed consent for data collection: Include the committee's ethical clearance date and number. Explain the process by which informed consent was approved. Written or verbal?

2. Line 115-119 : Study size :The study included all the (120) MHCPs of interest at MTRH.

Selection of participants: Consecutive sampling method was used until all eligible participants in the study are enrolled.

Please identify how did you choose your sample size? How many care givers are there in your institute? Did you asked all of them?

Reviewer #2: I wondered how the authors got an equal sex distribution of their respondents. Was this by chance or deliberate? This was not clearly stated in the methodology. This can be found in the result section under the sociodemographic characteristics of maternal healthcare providers (table 1).

6. PLOS authors have the option to publish the peer review history of their article (what does this mean?). If published, this will include your full peer review and any attached files.

Reviewer #1: **Yes: **SHAHLA KAREEM ALALAF

Reviewer #2: **Yes: **Dr. Habiba Ibrahim Abdullahi

---

## [Author Response · Author response to Decision Letter 0]

10 Nov 2023

Reviewer #1:

Title Concerns: We appreciate the reviewer's concern about the title's scope. We acknowledge that the study was conducted at a single institution, but the intention was to shed light on the labor pain relief practices in this tertiary facility, which often serves as a reference point for medical care in the region. We understand the concern about generalizing to all of Kenya and agree that it may be misleading. Therefore, we have revised the title to "Labour Pain Relief Practice by Maternal Health Care Providers at a Tertiary Facility in Kenya: An Institution-Based Descriptive Survey" to more accurately reflect the study's scope.

Ethical Clearance and Informed Consent: We thank the reviewer for pointing out the need for additional information regarding the ethical clearance and informed consent process. We have updated the manuscript to include the Moi University School of Medicine/MTRH Institutional Research and Ethics Committee's ethical clearance date and number. Additionally, we have clarified that written informed consent was obtained from all participants.

Sample Size and Selection: The reviewer rightly questioned the methodology for sample size determination and participant selection. We have added a brief explanation regarding this. ”The study employed a census that included all the 120 Health care providers of interest at MTRH”. As for participant selection, we used of consecutive sampling to enroll all eligible participants in the study.

Reviewer #2:

The reviewer pointed out an important question about the equal sex distribution of respondents in the methodology. It was coincidental that there was an equal number of male and female participants as the study included all maternal health care providers at the unit.

---

## [Decision Letter · Decision Letter 1]

7 Feb 2024

Labour pain relief practice by maternal health care providers at a Tertiary Facility in Kenya: An institution-based descriptive survey.

PONE-D-23-09684R1

Dear Kimbley Asaso Omwodo,

We’re pleased to inform you that your manuscript has been judged scientifically suitable for publication and will be formally accepted for publication once it meets all outstanding technical requirements.

Kind regards,

Adu Appiah-Kubi, MBChB, CEMBA, FGCS

Academic Editor

PLOS ONE

Additional Editor Comments (optional):

Reviewers' comments:

Reviewer's Responses to Questions

**Comments to the Author**

1. If the authors have adequately addressed your comments raised in a previous round of review and you feel that this manuscript is now acceptable for publication, you may indicate that here to bypass the “Comments to the Author” section, enter your conflict of interest statement in the “Confidential to Editor” section, and submit your "Accept" recommendation.

Reviewer #1: All comments have been addressed

2. Is the manuscript technically sound, and do the data support the conclusions?

Reviewer #1: Yes

3. Has the statistical analysis been performed appropriately and rigorously? 

Reviewer #1: Yes

4. Have the authors made all data underlying the findings in their manuscript fully available?

Reviewer #1: Yes

5. Is the manuscript presented in an intelligible fashion and written in standard English?

Reviewer #1: Yes

6. Review Comments to the Author

Reviewer #1: The authors have addressed adequately addressed my earlier comments. The article is suitable for publication now

7. PLOS authors have the option to publish the peer review history of their article (what does this mean?). If published, this will include your full peer review and any attached files.

Reviewer #1: **Yes: **PROFESSOR SHAHLA KAREEM ALALAF

---

## [Editor Report · Acceptance letter]

28 Feb 2024

PONE-D-23-09684R1 

PLOS ONE

Dear Dr. Omwodo, 

I'm pleased to inform you that your manuscript has been deemed suitable for publication in PLOS ONE. Congratulations! Your manuscript is now being handed over to our production team.

Kind regards, 

on behalf of

Dr. Adu Appiah-Kubi 

Academic Editor

PLOS ONE